# Commonsense-T2I Challenge: Can Text-to-Image Generation Models Understand Commonsense?

**Xingyu Fu[¶][*], Muyu He[¶][*], Yujie Lu[§][*], William Yang Wang[§], Dan Roth[¶]**
University of Pensylvania[¶], University of California, Santa Barbara[§]
{xingyuf2, muyuhe, danroth}@seas.upenn.edu, {yujielu, wangwilliamyang}@ucsb.edu
https://zeyofu.github.io/CommonsenseT2I

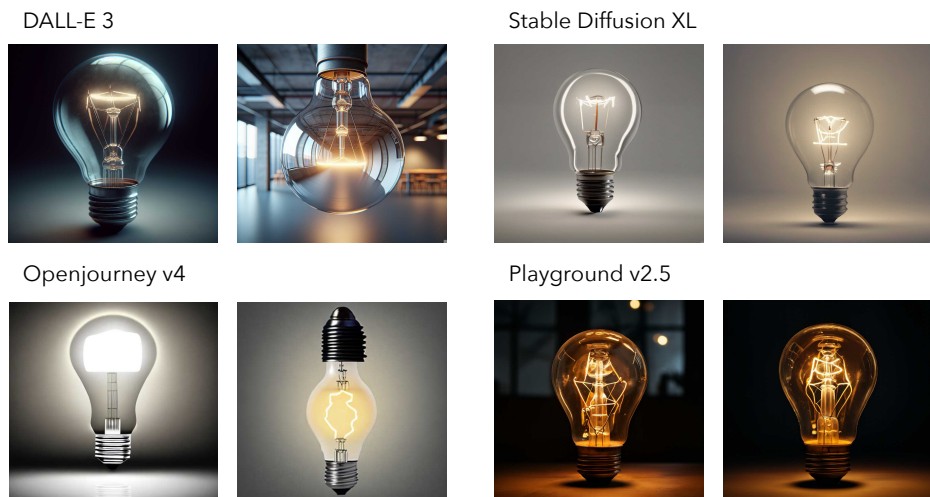

*prompt: A lightbulb without electricity*

Figure 1: An example prompt in `Commonsense-T2I` and failure cases from DALL-E 3 (Betker et al., 2023), Stable Diffusion XL (Rombach et al., 2022), Openjourney v4, and Playground v2.5 (Li et al., 2024). The expected output for the prompt is "`The lightbulb is unlit`".

## Abstract

We present a novel task and benchmark for evaluating the ability of text-to-image(T2I) generation models to produce images that align with commonsense in real life, which we call `Commonsense-T2I`. Given two adversarial text prompts containing an identical set of action words with minor differences, such as "`a lightbulb without electricity`" *vs.* "`a lightbulb with electricity`", we evaluate whether T2I models can conduct visual-commonsense reasoning, *e.g.* produce images that fit "`The lightbulb is unlit`" *vs.* "`The lightbulb is lit`" correspondingly. `Commonsense-T2I` presents an adversarial challenge, providing pairwise text prompts along with expected outputs. The dataset is carefully hand-curated by experts and annotated with fine-grained labels, such as commonsense type and likelihood of the expected outputs, to assist analyzing model behavior. We benchmark a variety of state-of-the-art (sota) T2I models and surprisingly find that, there is still a large gap between image synthesis and real life photos–even the DALL-E 3 model could only achieve 48.92% on `Commonsense-T2I`, and the stable diffusion XL model only achieves 24.92% accuracy. Our experiments show that GPT-enriched prompts cannot solve this challenge, and we include a detailed analysis about possible reasons for such deficiency. We aim for `Commonsense-T2I` to serve as a high-quality eval-

---

[*] These authors contributed equally to this work.

uation benchmark for T2I commonsense checking, fostering advancements in real life image generation.

# 1 Introduction

Recent advances in generative modeling have allowed text-to-image (T2I) synthesis to achieve drastic performance improvements (Ramesh et al., 2021; 2022; Rombach et al., 2022; Betker et al., 2023; Li et al., 2024). While it seems that we can realize a complete transition between a text prompt and an image, an image is worth a thousand words, and it is inevitable to lose information between the transition, due to the simplicity nature of language. Therefore, it is not possible to provide a text prompt to a T2I model that covers every single detail about an image–the T2I model must try to understand the text prompt first and self-imagine a real life scenario that contains every missing piece of detail, before generating the required images. As shown in Figure 1, the visual differences of "lightbulb" between "A lightbulb without electricity" and "A lightbulb with electricity" are blatantly obvious – but will the same still be true for machines? At least current T2I models Betker et al. (2023); Rombach et al. (2022); Li et al. (2024) fail to reason the commonsense for this example, namely "lightbulb is unlit without electricity".

We observe that many samples in existing T2I generation evaluations are straightforward composition of objects and their attributes, *e.g.* "A red car and a white sheep" (Saharia et al., 2022; Park et al., 2021; Cho et al., 2023; Feng et al., 2022). They focus on understanding of object-related and attribute-related tokens in the prompts, such as size, color, object, and shape, and fail to cover complicated commonsense reasoning required scenarios. Therefore, it remains unknown whether current generative AI systems can reach human-level intelligence and generate images that align with commonsense in reality, using existing evaluations.

In this paper, we specifically focus on the following question: can text-to-image models generate images that align with commonsense in reality? Motivated by this, we propose a novel evaluation task, called Commonsense-T2I, for measuring commonsense reasoning capabilities in generative models. As illustrated in Figure 2, Commonsense-T2I presents a high-quality expert-curated test set, with each data sample containing two adversarial text prompts, their corresponding expected output descriptions, likelihood score for each expected output, and commonsense category. We design the prompts in a pairwise format that both prompts contain an identical set of action words with minor differences, ordered in such a way that the images must show noticeable differences to align with commonsense in real life. To perform well on Commonsense-T2I, T2I models must not only encode the superficial meaning of each token in the prompts, but also be able to synthesize commonsense reasoning across the two modalities.

One natural question is how to conduct commonsense reasoning assessment on T2I models. Most metrics evaluate the models on image-text alignment (Radford et al., 2021; Hessel et al., 2021), focusing on divers subjects including fidelity (Heusel et al., 2017; Jayasumana et al., 2023), faithfulness (Hu et al., 2023), and compositionality (Lu et al., 2023c; Chen, 2023). None of them examines commonsense reasoning in generative models. While some recent methods evaluate models by human feedback (Lu et al., 2024), they are effort-demanding, do not focus on commonsense, and only include relative comparison without gold answers. In this paper, we present an automatic evaluation pipeline using our collected expected output descriptions and multimodal large language models (LLMs) (OpenAI, 2023; Team et al., 2023), that is tested to align well with human perceptions under our evaluation metric.

The primary purpose of Commonsense-T2I is to support commonsense probing tasks for T2I models. We experiment with a variety of generative models including Stable Diffusion models, Playground v2.5, Openjourney v4, and DALL-E 3(Rombach et al., 2022; Betker et al., 2023; Li et al., 2024). Surprisingly, the state of the art (sota) DALL-E model 3 only achieves 48.92% accuracy, and all other models hover around 15-30% accuracy. Our findings indicate that the commonsense reasoning capabilities have not emerged in existing T2I models. Additional experiments(§3.3) show that GPT-revised enriched prompts cannot

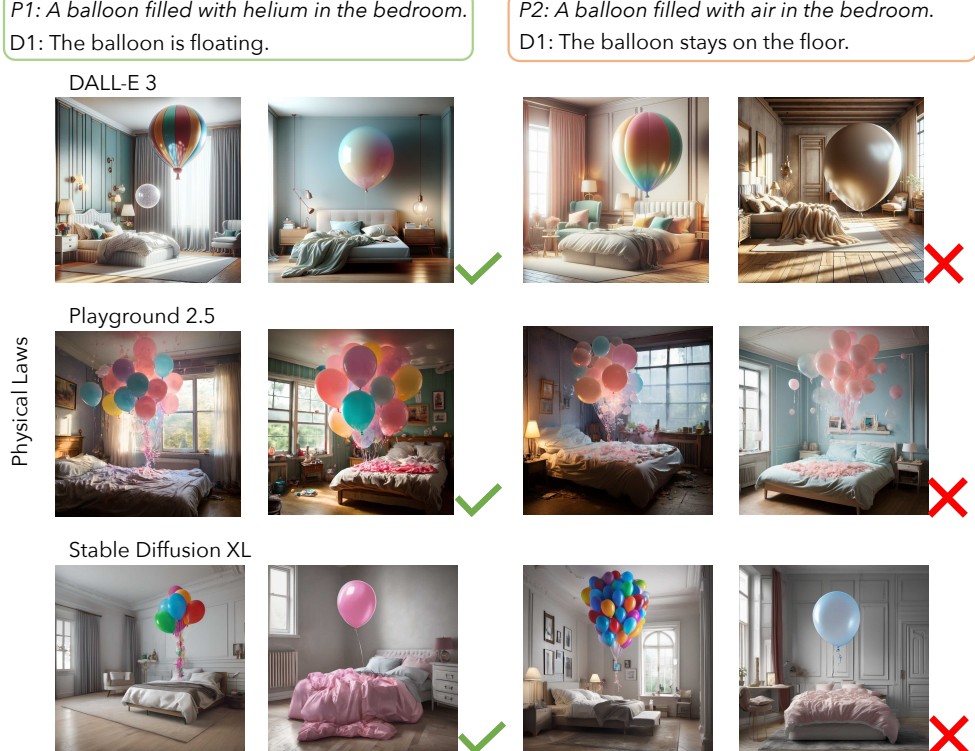

Figure 2: Illustration of one data example from Commonsense-T2I, where $P_1, P_2$ are pairwise prompts and $D_1, D_2$ are expected output description(§2), along with selected generated images from DALL-E 3, Playground v2.5, and Stable Diffusion XL. More examples in §3.3

solve the Commonsense-T2I challenge, and we include detailed analysis on possible reasons for such deficiency across all the T2I models.

In summary, our contributions are threefold. (1) We propose a high-quality expert-annotated benchmark for evaluating commonsense reasoning required in text-to-image generation. (2) We propose an automatic evaluation pipeline using multimodal LLMs for the task, and show that it is highly correlated with human evaluation. (3) We benchmark a wide range of T2I models on Commonsense-T2I and show that there is still a huge gap between all current models and human level intelligence, along with detailed analyses. We hope that Commonsense-T2I will stimulate the community to help T2I models catch up with human-level commonsense reasoning capabilities, fostering further progress in the field.

## 2 The Commonsense-T2I Benchmark

Our goal is to faithfully evaluate whether T2I models understand commonsense. We introduce a novel benchmark, Commonsense-T2I, designed to enable both quantitative and qualitative evaluation of the real-life commonsense reasoning capabilities of T2I models. We unfold this section by illustrating the overall design of Commonsense-T2I (§2.1) and discussing its unique features. Then we provide an in-depth explanation of the data curation process (§2.2). We further present the evaluation metric designed for the task(§2.3).

### 2.1 Overview of Commonsense-T2I

Commonsense-T2I comprises of 150 manually curated examples, where each example has a pair of adversarial prompts: $P_1$ and $P_2$, their corresponding expected output descriptions: $D_1$ and $D_2$, likelihood scores for each output to happen, and commonsense category. Complete data example is in A.1. A data sample satisfies the Commonsense-T2I criteria if and only if:

- $P_1$ and $P_2$ have the same set of action words but different subjects or adjectives.
- $P_1$ and $P_2$ have the same subjects but different action words.
- $D_1$ and $D_2$ are completely contrastive and cannot co-exist in one image.
- $P_1$ will lead to $D_1$ and $P_2$ will lead to $D_2$ in daily life under common sense.

| Category | Examples | Percent |
|---|---|---|
| Physical Laws | A glass of water fell on the floor: spilled water.
A hot cup of water in winter: steam rises. | 32.7 |
| Human Practices | A man at a wedding: in suit, looking cheerful.
A person eating a burger: eating with hands. | 30.0 |
| Biological Laws | Oak trees during winter: no leaves on branches.
A wheat field in spring: green field. | 11.3 |
| Daily Items | A small bag of party balloons for sale: balloons are flat.
A phone with a drained battery: dark screen. | 14.0 |
| Animal Behaviors | A peacock attracting a mate: spreading feathers.
A penguin sliding on ice: sliding on its belly. | 12.0 |

Table 1: Commonsense knowledge categories and percentage of data samples.

## 2.2 Dataset Collection Process

The Commonsense-T2I dataset is entirely hand-curated by experts. We first decide on the categories of commonsense knowledge required for text-to-image generation, and then use GPT-4 (Brown et al., 2020) to help generate multiple examples requiring visual-commonsense knowledge as inspirations, and manually curate each test sample of pairwise prompts and expected outputs.

**Commonsense Category**. Commonsense knowledge is a huge portion of human experience, encompassing knowledge about the spatial, physical, social, temporal, and psychological aspects of typical everyday life (Liu & Singh, 2004), and widely studied in AI research (Bosselut et al., 2019; Zellers et al., 2019). To build Commonsense-T2I, we manually select five categories of knowledge that naturally require visual commonsense understanding, as illustrated in Table 1.

**Inspiration Generation**. We prompt the GPT-4-turbo model iteratively to generate a massive pool of specific examples as inspirations for each commonsense category. Specifically, we first use GPT-4 to generate natural sentence examples given a commonsense category, *e.g.* "butter melts when heating" given category *physical laws*. Additionally, we require GPT to only provide examples that are visually salient and easy to visualize. For instance, "a bowl of milk put outside the fridge for a few days" is a bad case because it entails a non-visual commonsense knowledge that milk would turn sour. For each example, we prompt GPT-4 to locate the subject and expected output, and discover specific real scenarios as $P_1$, *e.g.* the subject is "*butter*", scenario is "*in a heated pan*", and expected output is "*butter melts in a heated pan*". Finally, we ask GPT-4 to generate one counter-example for each real scenario with a slightly different subject or different action for the same subject, as the temporary candidate for $P_2$. However, the quality of generated examples are often not guaranteed, and may not test the desired commonsense knowledge.

**Manual Curation**. Prepared with the auto-generated examples as inspirations, we manually create the pairwise prompts and expected output descriptions, and verify the commonsense type, add the likelihood score. For the pairwise prompts, we rewrite to have natural sentences that do not reveal the expected outputs. For instance, we turn "an untied air balloon" into "A balloon filled with air" as in Figure 2. Also, we revise to keep minimum difference between $P_1$ and $P_2$, while leading to contrasting outputs per criteria. For the expected output descriptions, we rewrite to be faithful to the prompts. For example, we changed the GPT output "*A mirror reflecting nothing in the room*" to "*A barely visible room*" for "A mirror in a room without light". For the likelihood score, we rate to answer "How

many out of ten generated images do we believe should show the expected outputs?", and discard examples with likelihood score lower than 7.

**Data quality control:** To guarantee the quality of `Commonsense-T2I`, we manually go through all collected data again and filter out data that are ambiguous.

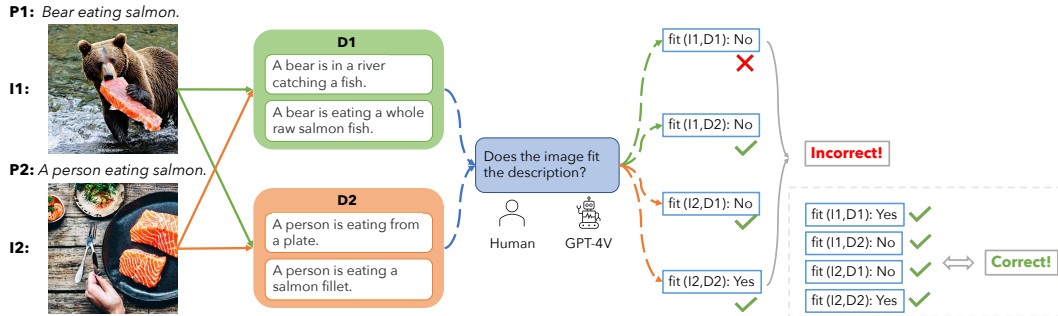

Figure 3: The evaluation pipeline for `Commonsense-T2I`. More details are in Section 2.3.

## 2.3 Evaluation Metrics

`Commonsense-T2I` is designed as a pairwise challenge set and we evaluate performance of T2I models according to the following criteria: only when both of the pairwise prompts $P_1$ and $P_2$ have the generated images $I_1$ and $I_2$ match the expected output descriptions $D_1$ and $D_2$ at the same time, we count the sample as correct. Specifically,

$$fit(I_i\_D_j) = \begin{cases} 1, & \text{if image } I_i \text{ fits the description } D_j \\ 0, & \text{otherwise} \end{cases}$$

is an indicator function evaluating whether the generated image $I_i$ fits the description $D_j$ where $i, j \in [1, 2]$. Then, the score for data sample $n$ containing pairwise prompts $P_1^n$ and $P_2^n$ is calculated as

$$score_n = \begin{cases} 1, & \text{if } fit(I_1^n\_D_1^n) + fit(I_2^n\_D_2^n) - fit(I_1^n\_D_2^n) - fit(I_2^n\_D_1^n) = 2 \\ 0, & \text{otherwise} \end{cases}$$

where it is only correct when the generated images for the pairwise prompts are both correct at the same time. For instance, if the T2I model generates images for $P_1$ correctly but images for $P_2$ incorrectly, namely $fit(I_1\_D_1) = 1$ and $fit(I_2\_D_2) = 0$, then we consider the sample as incorrect, because the model fails to conduct the required commonsense reasoning. In our experiments, we generate multiple times for each data sample and take the average score for fair comparison. The final accuracy is then calculated as

$$Accuracy = \frac{1}{N} \sum_{n=1}^{N} score_n$$

where N is the total number of data samples. Details can be found in Figure 3.

## 3 Experiment

In this section, we first describe the baseline T2I models and experimental setup (§3.1). Then we present a comprehensive evaluation of both human and existing multimodal models (§3.2). We demonstrate that while T2I models can generate high-quality images, `Commonsense-T2I` is challenging for existing models. Finally, we provide detailed analyses on using multimodal large language models (LLMs) as auto evaluators, whether GPT-enriched prompts can solve the problem, possible reasons for dificiency in current models, and error analysis across different T2I models(§3.3).

### 3.1 Experimental Setup

**Baseline T2I Models:** We evaluate `Commonsense-T2I` on a variery of T2I models, including three Stable Diffusion (Rombach et al., 2022) models: (1) Stable Diffusion v2.1[1] (SD-21), (2) Stable Diffusion XL[2] (SD-XL), and (3) Stable Diffusion 3 Medium (Esser et al., 2024) (SD-3). For the models SD-XL and SD-3, we include a new setting that uses pairwise prompts as negative prompts in input: (4) SD-XL w/ negative prompt and (5) SD-3 w/ negative prompt. Developed upon the Stable Diffusion XL model, (6) Playground v2.5 (Li et al., 2024)[3] is included as it provides high-quality images that are preferred by human over Stable Diffusion XL and DALL-E 3 (Betker et al., 2023) as in the paper. We also include (7) Openjourney v4[4] from PromptHero[5], which is finetuned upon Stable Diffusion v1.5 using Midjourney[6] images, and (8) the Latent Consistency Models (LCMs) (Luo et al., 2023) [7], which is distilled from Dreamshaper v7[8] fine-tune of Stable Diffusion v1.5. Flux models developed by Black Forest Labs[9] are recently released and have shown great performances. We include (9) Flux Dev and (10) Flux Schenel for comparison. As for API-based models, we evaluate on the DALL-E 3 model (Betker et al., 2023). The DALL-E 3 model by default enriches and revises the given text prompt using GPT model (Brown et al., 2020) before generation, adding more details to the prompt since more detailed prompts generally result in higher quality images. Therefore, we include two variants of the model: (11) DALL-E 3, which is the original model, and (12) DALL-E 3 w/o revision, which turns off the GPT-revision function–we follow the OpenAI instruction[10] and add the following to our prompts: `I NEED to test how the tool works with extremely simple prompts. DO NOT add any detail, just use it AS-IS: {prompt}.`

**Evaluation Protocol:** We assign two experts (coauthors) for each data sample in `Commonsense-T2I` and present their average scores as human performance. As stated in Section 2.3, evaluators are expected to separately determine whether the content in image $I_1^n$ fits the description $D_1^n$, and also whether image $I_2^n$ fits the description $D_2^n$, for data sample $n$. In order to conduct fair comparison and avoid randomness, we generate the images for each data sample four times, evaluate separately, and take the average score for each data sample to calculate final accuracy. Namely, $score_n$ is averaged over four times of generations.

Notice that due to the demanding nature of the task, we have conducted manual evaluations only on the following models: SD-21, SD-XL, DALL-E 3, and DALL-E 3 w/o revision.

**Automatic Evaluation:** Considering the effort-demanding nature of human evaluation, we experiment with multimodal LLMs to conduct automatic evaluation on `Commonsense-T2I`. Two models are tested in our paper: GPT-4V(ision) (OpenAI, 2023), which is known to be one of the most powerful multimodal models to date, and we evaluate using three checkpoint models: gpt-4-vision-preview, gpt-4-turbo, and gpt-4o; and GeminiPro (Team et al., 2023), which is one of the most widely used multimodal models, and we use the Gemini 1.0 Pro Vision version of it. Specifically, for each image $I$ and description $D$, we get $fit(I\_D)$ with the following prompt: `Can you tell me if the image generally fits the descriptions "{description}"? If it generally fits the descriptions, then return 1, otherwise, return 0. Give me number 1 or 0 only.` Notice that in several cases where the multimodal LLM fails to tell $fit(I\_D)$ correctly and believes the image fits

---

[1] https://huggingface.co/stabilityai/stable-diffusion-2-1

[2] https://huggingface.co/docs/diffusers/en/using-diffusers/sdxl

[3] https://huggingface.co/playgroundai/playground-v2.5-1024px-aesthetic

[4] https://huggingface.co/prompthero/openjourney-v4

[5] https://prompthero.com/

[6] https://www.midjourney.com/home

[7] https://huggingface.co/SimianLuo/LCM_Dreamshaper_v7

[8] https://huggingface.co/Lykon/dreamshaper-7

[9] https://blackforestlabs.ai/

[10] https://platform.openai.com/docs/guides/images/prompting

| Model/Evaluator | CLIP (Radford et al., 2021) | Gemini Pro (Team et al., 2023) | GPT-4 preview (OpenAI, 2023) | GPT-4 Turbo (OpenAI, 2023) | GPT-4o (OpenAI, 2023) | Human |
|---|---|---|---|---|---|---|
| *Open-source T2I models* | | | | | | |
| SD-21 (Rombach et al., 2022) | 24.67 | 21.67 | 15.83 | 21.17 | 21.00 | 18.83 |
| SD-XL (Rombach et al., 2022) | 26.00 | 29.67 | 23.50 | 26.67 | 25.17 | 24.92 |
| SD-XL w/ negative prompt | 44.83 | - | 23.50 | 30.67 | 31.67 | - |
| Openjourney v4 | 26.83 | 26.67 | 18.83 | 20.17 | 22.33 | – |
| Playground v2.5 (Li et al., 2024) | 24.83 | 29.00 | 18.50 | 20.33 | 25.83 | – |
| LCMs (Luo et al., 2023) | 23.33 | 23.50 | 17.17 | 20.67 | 23.83 | – |
| SD-3 (Rombach et al., 2022) | 26.17 | - | 22.00 | 22.83 | 21.67 | - |
| SD-3 w/ negative prompt | 47.17 | - | 30.00 | 35.83 | 37.67 | - |
| Flux Dev | 24.50 | - | 24.33 | 23.83 | 19.00 | - |
| Flux Schenel | 27.50 | - | 29.67 | 26.17 | 28.00 | - |
| *proprietary T2I models* | | | | | | |
| DALL-E 3 (Betker et al., 2023) | **40.17** | **45.50** | **43.83** | **45.83** | **48.83** | **48.92** |
| DALL-E 3 w/o revision | 34.83 | 36.50 | 33.50 | 35.00 | 37.50 | 34.00 |

Table 2: **Main results on the `Commonsense-T2I` challenge set**. The columns row shows the T2I models that we evaluate on, and the first row shows the evaluator choices. The best performance model under each evaluator is in-bold. Notice that some Gemini 1.0 Pro Vision scores are left blank because the model was deprecated since July 2024.

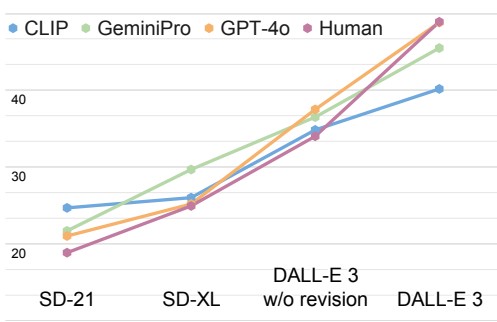

Figure 4: Comparison between using multi-modal LLMs *vs.* humans as evaluators. We can see that the evaluators CLIP and GeminiPro are both not ideal. GPT-4o, however, consistently provides similar evaluation scores to humans and can be a good candidate for automatic evaluation.

Figure 5: Illustration of the CLIP embedding similarity of prompts $P_1$, $P_2$ against human evaluated performance scores. It suggests that T2I models perform badly when their text encoders fail to differentiate between $P_1$ and $P_2$, and perform well when $P_1$ and $P_2$ are correctly embedded to be far.

both expected output descriptions $D_1$ and $D_2$, we randomly select one description such that $fit(I\_D_1) + fit(I\_D_2) = 1$.

We also include CLIP (Radford et al., 2021) (ViT-L/14) as an additional evaluator, considering that it is one of the most widely used multimodal encoders. Specifically, we give CLIP the generated image $I$ and two expected output descriptions $D_1$ and $D_2$, and let $fit(I\_D_1) = 1, fit(I\_D_2) = 0$ if the cosine similarity between $I$ and $D_1$ is larger than that between $I$ and $D_2$, or $fit(I\_D_1) = 0, fit(I\_D_2) = 1$ if vice versa.

### 3.2 Main Results

**Human Evalution Results:** As illustrated in Table 2, the mean accuracy of open-source stable diffusion models hover below 25%, with the XL model achieving a 6.09% improvement over the v2.1 model. We surprisingly find that even the sota DALL-E 3 model only achieves 48.92%, meaning that it fails on the commonsense challenge for more than half

of the cases in `Commonsense-T2I`. After turning off the GPT revision function of DALL-E 3, DALL-E 3 w/o revision achieves 15.92% worse than the DALL-E 3 performance, suggesting the possible improvements brought by enriched text prompts.

**Automatic Evaluation Results:** Can Multimodal LLMs replace human evaluators? As shown in Table 2, with the most advanced models, GPT-4o and Gemini Pro, we can achieve automatic evaluation performances similar to that of humans, with the maximum difference being 5.09% and 5.25% respectively. Their performance trends are consistent with that of humans, as shown in Figure 4. Notably, GPT-4V consistently rates lower than humans, while Gemini Pro always rates higher than humans except on DALL-E 3 images. We believe that multimodal LLMs, especially GPT-4V, can represent the T2I model performances generally. However, CLIP evaluation scores are relatively divergent from that of human, and cannot represent the trends of T2I model performances well.

## 3.3 Analysis

**Are T2I models limited by text embedding?** Since all the Stable Diffusion (based) T2I models score under 35% accuracy on `Commonsense-T2I`, we investigate the possible reason behind this phenomena: these models might be biased by the text embedding of the prompts. The motivation is follows: if the embeddings of $P_1$ and $P_2$, which are inputs to the T2I models, are very similar, then they could lead the T2I models to generate similar images for $P_1$ and $P_2$, while the expected outputs should different. We deploy the CLIP (Radford et al., 2021) (ViT/L14) encoder, which is the default text encoder for Stable Diffusion (based) models, to encode the pairwise prompts $P_1$ and $P_2$ in `Commonsense-T2I`. We compare the similarity between CLIP embedding of $P_1$ and $P_2$ against performance score as in Figure 5. Notice that we adopt min-max normalization to project the embedding similarity values into [0,1].

**Can GPT augmented prompts solve the `Commonsense-T2I` problems?** To answer this question, we analyze the error cases of DALL-E 3, which automatically uses GPT-augmented revised-prompts, and check whether the revised prompts include correct expected outputs. As in Figure 6, we show the difference between DALL-E 3 outputs and DALL-E 3 w/o revision outputs, along with the GPT-revised prompt used by DALL-E 3. We can see that in the first two failed cases, the GPT revised prompts can provide the exact correct expected output information; while in the last two cases, they provide partially correct information. In short, GPT augmented prompts can help to some extent, with DALL-E 3 achieving 14.92% improvement over DALL-E 3 w/o revision. However, they cannot solve the `Commonsense-T2I` challenge – they either fail to provide comprehensive correct details, or the T2I part fails to visualize the correct details.

**Do different T2I models make same errors?** The Stable Diffusion based models: SD-21, SD-XL, Playground v2.5, and Openjourney v4 fail on most samples in `Commonsense-T2I`, even for the easy ones, such as "`prompt: A peacock sleeping, expected: The peacock has its feathers closed`" from "`Animal Behaviors`". Additional failed examples are shown in Figure 9. Meanwhile, DALL-E 3, as illustrated in Figure 6, often fails on more complicated cases, *e.g.* it mostly succeeds on "`Animal Behaviors`" and "`Biological Laws`" samples and fails on *uncommon* situations with commonly seen objects, such as `unlit lightbulbs`, `distinguished candles`, `fully melted ice cream`,...,*etc*.

## 4 Related Work

**T2I Models and Benchmarks** Text-to-image synthesis models are typically trained to generate images conditioned on text. Early studies widely use GANs (Reed et al., 2016b;a). Starting from DALL·E (Ramesh et al., 2021), image generation models started to show impressive results. Recently, diffusion models have achieved remarkable success on text-guided image generation (Nichol et al., 2021; Saharia et al., 2022; Ramesh et al., 2022; Rombach et al., 2022; Li et al., 2024; Ku et al., 2024). Another series of work (Lu et al., 2023b; Zhu et al., 2023) enables the collaboration between the LLMs and text-to-image models. Multiple benchmarks evaluating different aspects of T2I models have been introduced.

DALL-E 3

*prompt: A lightbulb without electricity*
*expected: Lightbulb is unlit*

*revised prompt*: Visualize a clear glass lightbulb. The filament inside is untouched and frail, ==but there's no glow emanating from it.== The metallic base is screwed tightly, waiting for the moment to be set in a power source. However, currently, it's devoid of electricity. Around it, the surroundings are dark and still, amplifying its ghostly inert state. The lightbulb stands as a testament of technology, yet it's symbolizing a paradoxical illustration of darkness in this instance. Please show it prominently, taking center stage in this image.

DALL-E 3 w/o revision

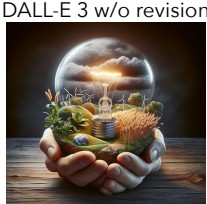

DALL-E 3

*prompt: a pen placed in a cup of water*
*expected:* The submerged part of the pen appears bent

*revised prompt*: An image depicting a realistic scenario where a ballpoint pen is immersed in a clear glass cup filled with water. The cup should be resting on a flat and clean surface, and light should be reflecting off the water and the glass, giving them a glossy look. ==The pen inside the water is distorted due to the refraction of light.== Shadows from pen and the cup are visible on the surface.

DALL-E 3 w/o revision

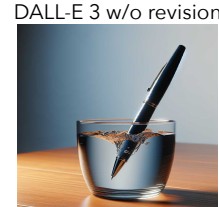

DALL-E 3

*prompt: an hourglass just finishing its counting*
*expected: Sa*nd is mostly in the bottom part

*revised prompt*: An ultra realistic image of a vintage hourglass right at the end of its countdown, ==the last grain of sand dropping from the top chamber to the bottom one.== The hourglass has an ornate design and is crafted from dark polished wood and glistening, clear glass. The fine grains of sand within it are golden in color. It stands on a richly varnished wooden table and behind it is a nearly fading neutral background.

DALL-E 3 w/o revision

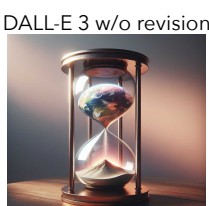

DALL-E 3

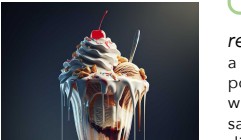

*prompt: a sundae untouched for several hours*
*expected:* The ice cream has melted into liquid form

*revised prompt*: Generate a detailed and hyper realistic image of a sundae that has been left untouched for several hours. It should portray ==clear signs of melting and decomposition of its ingredients== which typically includes whipped cream, ice cream of various flavors, sauces like chocolate and caramel, nuts, cherries, and a glass serving dish. There should be visible signs of the condensation on the outer surface of the dish. The background can include a table surface with some splatters and drops of melted ice cream and ingredients.

DALL-E 3 w/o revision

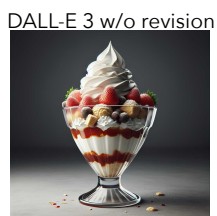

Figure 6: Error cases of DALL-E 3. Prompts and expected outputs are in the green box. DALL-E 3 images are generated with the revised prompts returned by DALL-E 3, and DALL-E 3 w/o revision images are generated with the original prompt. The highlighted sentences are (partially) correct expected output descriptions in revised prompts.

Most of them focus on straighforward text prompts, e.g. *"a read book and a yellow vase."*, and evaluate direct visual attributes such as counting, color, and shape (Saharia et al., 2022; Park et al., 2021). Others focus on object detection and relations (Cho et al., 2023; Bakr et al., 2023), compositionality: presence of multiple objects and their attributes (Bakr et al., 2023; Huang et al., 2023), and fairness (Cho et al., 2023; Bakr et al., 2023). But none of them evaluates multimodal commonsense understanding of generative models.

**T2I Evaluation**    Most metrics evaluate the models on fidelity (Salimans et al., 2016; Heusel et al., 2017; Jayasumana et al., 2023), image-text alignment (Radford et al., 2021; Hessel et al., 2021; Li et al., 2023), Recent metrics try to use large language models (LLMs) (Lu et al., 2023c; Zhang et al., 2023; Chen, 2023), or VQA (Hu et al., 2023), or human (Ku et al., 2024; Lu et al., 2024) for evaluation. However, there is no comprehensive study on how well those evaluation metrics work for commonsense T2I generation. We propose evaluation metrics

specifically designed for our task and validate that our proposed metrics align well with human perceptions.

**Multimodal Large Language Models**   With the recent development of multimodal Large Language Models (LLMs) (Alayrac et al., 2022; Li et al., 2023; Liu et al., 2023a; OpenAI, 2023; Liu et al., 2023b; Team et al., 2023; Zhang et al., 2023), more reasoning-related research questions about multimodality have been studied (Thrush et al., 2022; Marino et al., 2019; Fu et al., 2022; Lu et al., 2023a; Fu et al., 2023a; Gupta & Kembhavi, 2023; Surís et al., 2023; Fu et al., 2024; 2023b; Wang et al., 2024; Hu et al., 2024). One paper (Zhang et al., 2022) studies visual commonsense knowledge in pretrained multimodal models. Concurrent work (Meng et al., 2024) specifically evaluates on the physics phenomenon in real life in text-to-image generation. However, there are no comprehensive studies on alignment with general commonsense in reality for T2I models.

## 5   Conclusion

We introduce Commonsense-T2I, a novel task for evaluating commonsense reasoning abilities of T2I models. We provide a high-quality expert-annotated test set for the task including pairwise prompts and expected outputs. Our experiments show that current T2I models score between 15-50% on our dataset, fostering future research in this direction.

**Limitations**   We would like to emphasize that the size of Commonsense-T2I is limited by the need to manually revise all the samples (each including five entries) by experts. We propose Commonsense-T2I as a high-quality expert-curated test set and believe it can serve as a good evaluation benchmark for the goal of this paper. Nevertheless, using the inspiration generation method in §2.2, one can easily generate huge amount of weak-supervision data.

## Acknowledgements

We thank Wenhu Chen for the insightful discussions. This work was funded in part by ONR Contract N00014-23-1-2417, and supported by NSF grant IIS-2212433.

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

## A Appendix

### A.1 Complete Data Example

A complete data sample in Commonsense-T2I looks as the following:

```
P₁: A birthday cake after making a wish
P₂: A birthday cake before making a wish
D₁: The candles are extinguished
D₂: The candles on the cake are lit
category: Human Practices
likelihood:9
```

## A.2 Dataset Inspiration Generation Prompts

We illustrate the prompts we used to generate the data inspirations from GPT-4-turbo as following: Figure 7 and 8.

I want you to generate some prompts for image generation AI. These prompts require the AI to have common sense to infer visual information and depict it correctly.
For example:
    A lightbulb without electricity cannot give light - requires Ai to infer that the lightbulb is dark - category: physical laws - prompt: A lightbulb without electricity.
    A crosswalk with a red pedestrian signal means stop - requires Ai to infer that no one is crossing the streets because of the red light - category: cultural practices - prompt: A crowded crosswalk with a red pedestrian signal.
    A guitar with broken strings doesn't have all six complete strings - requires the AI to infer that at least one of the six strings on the guitar is broken and curled up - category: visual common sense -  prompt: A guitar with broken strings.
    Bats sleep during the day - requires the AI to infer that bats are not flying around in broad daylight - category: animal behavior - prompts: bats during the day.
Now generate five more examples in the above format, without introduction or filler words.

Figure 7: Prompts we use to generate data inspirations. (1/2)

For each of the examples above,
I want you to put them in this format:

{{"prompt1": "<object> + <visible scenario>", "prompt2": "<a contrasting prompt about the same subject and common sense>", "category": "<category {number}'s name>", "description1": "A + <visible state> + <object>", "description2": "A + <visible state> + <object>"}}

In this format, description1 is about prompt1, and description2 is about prompt2.

For example:
{"prompt1": "A lightbulb without electricity", "prompt2": "A lightbulb with electricity", "description1": "The lightbulb is dark.", "description2": "The lightbulb is lit."}

I want the language of the prompt to be concise and natural.
I want description1 and description2 to be an easily visualizable scenario.

Figure 8: Prompts we use to generate data inspirations. (2/2)

## A.3 Error Cases Examples

In Figure 9, we illustrate some error cases by SD-XL and Playground v2.5.

Figure 9: Error cases of SD-XL andPlayground v2.5. The prompt and expected output description are provided in green box for each example.

