# OpenReview forum: "Commonsense-T2I Challenge: Can Text-to-Image Generation Models Understand Commonsense?"
_colmweb.org/COLM/2024/Conference — COLM_

### Official Review · Reviewer_673E · 2024-04-23

**Rating:** 8
**Confidence:** 4
**Ethics Flag:** 1

**Summary:**

This paper proposes a benchmark for evaluating text-to-image models at commonsense implicature. For instance, given a prompt like "a lightbulb without electricity", an illustration should show a dark lightbulb, not one that's lit up.

The paper considers 150 manually selected prompts, grouped in pairs P1 and P2 that cannot cooccur together. Then, models can be scored by how well an image generation for P1 matches P1 and all the things it entails (versus P2). The paper considers humans as well as CLIP and image caption evaluators.

Overall, it seems that models struggle at this task, even when provided ultra-descriptive captions.

--
overall: seems like all reviews are positive, so should be pretty clearcut to accept this paper!

**Reasons To Accept:**

* This seems like an interesting benchmark that could be a useful diagnostic for the community. Many text-to-image model generations are a bit weird and different from how humans perceive the world and create art, and quantifying this gap seems useful.
* The experiments seem well done at least to this reviewer. In particular, it seems interesting that ultra-descriptive captions for DALL-E 3 help but do not fully solve the problem.

**Reasons To Reject:**

* I am still a fan of this paper, but I'm a bit iffy on some of these prompts in the digital art case. There are ostensibly multiple goals with digital art and representing the physical world accurately is just one of them (along with emphasizing key features). For instance, the melting ice cream sundae example in Figure 5 indeed shows something _in the process of being melted_, emphasizing the second part of the prompt. If there was a way to restrict these models to only generating naturalistic real-world images, maybe this could be avoided, but I think the lack of specification might be a challenge here. I'd guess that more realistic text-to-image models (trained more naturalistic images versus digital art) should do better on this task, which might not be intended.
* To this reviewer, it's not 100% clear whether commonsense is a great label for this task as opposed to just simply entailment (like SNLI-style).

---

> ### Author Rebuttal · Authors · 2024-05-31
>
> Thank you for your insightful comments and valuable suggestions.
>
> **Digital Art concerns**
>
> It is a great point and worth further exploration. For generality, we used the same prompt for every model. However, certain T2I models like Dall E 3 tend to generate digital art images. Specifically for the melting-ice example, we tried the Dall E 3 model again and changed the prompt into “Give me a realistic image, a real world photo of ” + original prompt. While the generated images look more real, they still make the same mistake as in Figure 5. We leave this question to future research.
>
> **Commonsense concern**
>
> It is a great point that these examples might be summarized as entailment problems. However, considering that all reasoning questions can be summarized as entailment questions, we argue that commonsense might be a better title here.

---

### Official Review · Reviewer_4uPG · 2024-05-06

**Rating:** 8
**Confidence:** 4
**Ethics Flag:** 1

**Summary:**

The paper aims to evaluate the capability of text-to-image (T2I) generation models on understanding commonsense. The paper first collects a new manual-curated dataset as a new benchmark for evaluation. The paper also proposes a manual and an automatic pipeline to evaluate T2I model performance on the new dataset. The paper compares a few T2I models including DALL-E, Openjourney v4 and more.

**Questions To Authors:**

Please see "Reasons to Reject"

1. How were likelihood scores used eventually? I.e. why they were necessary?

2. Page 6 Evaluation Protocol, how was the disagreement between two experts resolved?

3. It will be interesting to know how to enrich/enlarge this benchmark in the future and maintain good quality.

**Reasons To Accept:**

The paper is overall well-written with clear motivation.

The paper collects a useful new benchmark for the community to evaluate the capability of T2I models.

The paper proposes an automatic pipeline to evaluate T2I capability on understanding commonsense, which is correlated with the manual evaluation process.

The paper concludes with a clear direction for future works to improve T2I models.

**Reasons To Reject:**

Some details in the methodology are missing.

1. Section 2.2 Page 4, it is unclear how the likelihood score was assigned based on what criteria.

2. Section 2.2 Page 4, how was the data quality control implemented? How to decide data is ambiguous?

---

> ### Author Rebuttal · Authors · 2024-05-31
>
> Thank you for your insightful comments and valuable suggestions.
>
> **likelihood score and ambiguity removal**
>
> The likelihood score was only necessary and used during our curation process, for filtering out ambiguous examples. Specifically, it represents “At least how many out of ten generated images should show the expected outputs” for the given prompt. For example, human experts believe at least 5 out of 10 generated images for prompt (“people holding an umbrella in a rainstorm”) should fit the expected output (“umbrella is inside out”), then the likelihood score is 5. Since each data sample has two prompts, we assign the lower likelihood score of the prompts to be the score for the sample. According to our criteria that samples with likelihood lower than 7 are ambiguous and should be discarded, this example will then be removed.
>
> We only keep the likelihood scores in the benchmark for completion purposes.
>
> **Data Quality Control**
>
> The human experts (authors) went through three iterations of data curation together for better data quality, focusing on multiple aspects including but not limited to: prompt quality (to focus on visual commonsense cases), ambiguous case removal (rating likelihood scores), data category balancing (to find more examples of certain categories), evaluation descriptions quality (to be succinct), etc.
>
>
> **Evaluation detail**
>
> We take the average score of two experts for each image, and take the average score for the four images of each data sample to calculate final accuracy. The average agreement between human experts over four manually evaluated models is 92%.
>
> **Enlarge this benchmark**
>
> It is a great question. During our curation process, we find that it is very difficult to automatically generate high quality data samples – the generated prompts in most cases are not common sense required, but only complex descriptive sentences. We leave this question to future research.

---

### Official Review · Reviewer_XsVU · 2024-05-10

**Rating:** 7
**Confidence:** 3
**Ethics Flag:** 1

**Summary:**

This paper establishes a novel benchmark, named Commonsense-T2I, for evaluating the ability of text-to-image (T2I) models to understand commonsense in the text prompt. A dataset is created with the help of GPT4 and then manually filtered and annotated. Automatic and manual evaluation metrics are explored to analyze the performance of state-of-the-art (SOTA) T2I models. The paper shows that although SOTA diffusion-based T2I models can generate high-quality images, they fail to properly capture and generate the underlying commonsense expressed by the text input, which indicates a potential direction to improve text-to-image generation.

**Reasons To Accept:**

- Understanding the commonsense in text input and reflecting it in the generated image is an important capacity of text-to-image generation models. This paper proposes an interesting benchmark to analyze existing T2I models and provides insights for further improving text-to-image generation.

- The dataset collection process is clever and reasonable. The powerful GPT4 is used to propose inspirations for commonsense knowledge, followed by manual curation.

- The author first conducts manual evaluation and further explores using CLIP or multimodal LLMs for automatic evaluation. The conclusion suggests that multimodal LLMs can make the benchmark's evaluation process much more scalable.

- The paper provides detailed error cases of DALL-E 3, indicating that the SOTA T2I models have limitations in 1) understanding the commonsense expressed by the text and 2) correctly reflecting the straightforward "translated/extended commonsense" into pixels. The conclusions of the paper can provide helpful inspiration for future work in the T2I generation.

**Reasons To Reject:**

- It is unclear how the categories of commonsense knowledge are determined. If the five categories come from a referred paper, please clarify the source. Also, there is no explanation of how the ratios of the five categories are determined.

- The naming and explanation of D1 and D2 are vague and hard to understand -- the name "expected output descriptions" is kind of ambiguous and could mean that the descriptions are the output of some model. There is a typo at the top of Figure 2: The "D1" there should be "D2".

---

> ### Author Rebuttal · Authors · 2024-05-31
>
> Thank you for your insightful comments and valuable suggestions.
>
> **Commonsense Categories**
>
> The categories are summarized from observations during data curation preparation by the authors. Before we manually collect the benchmark, we observe daily life examples from the real-world that require visual commonsense but are extremely difficult for T2I models. We summarize the representative and most common cases to decide our commonsense knowledge categories. We will release all data, code and data curation pipeline for further scaling needs.
>
> The ratio in Table1 is the percentage of numbers of examples in our benchmark of each category.
>
> **Naming of D1 and D2**
>
> We apologize for the confusion. We intend to mean the descriptions of the output of tested T2I models. We will clarify the naming in our final version.
>
> Typo: Thanks for pointing it out. We will fix it in the final version.

---

### Official Review · Reviewer_FMFR · 2024-05-11

**Rating:** 7
**Confidence:** 4
**Ethics Flag:** 1

**Summary:**

The paper present a new benchmark called *Commonsense-T2I* challenge. The main contribution is that the author collected a set of text prompts involving commonsense and some of them are long-tail scenarios -- things won't happen often and they won't usually appear in the training corpus. The authors find that most common T2I models fail this challenge significantly. The best among all is DALLE3 and revision of prompts usually won't help -- even though the revision contains the correct commonsense reasonings.

**Reasons To Accept:**

- **Clear motivation**. The motivation of benchmarking T2I model's commonsense is welcome. The benchmark is also very needed given the conceptions that the common mass belief T2I is a solved problem but in fact it's not. With the introduction of commonsense T2I, people would see a new direction of improving T2I models.

- **Complete Ablation Study**. Though ablation study for datasets/benchmark papers are not fully required but it's nice to see the authors provide the ablation of different evaluator (automatic ones such as GPT-4 and Gemini, together with human raters). It's nice to see their consistencies and would confirm the benchmark is well designed.

**Reasons To Reject:**

1. I would recommend the authors expand the number of T2I models used for evaluation. Probably including recent latent consistency models and SD-3 to see if they have different biases against SD-21 and SD-XL.

2. I would love to hear more about the authors comments on potential ways to improve commonsense T2I. As the authors can collect these amount of benchmark data, the data can also be used to fine-tuning T2I models. For example, using fully-automated methods such as DreamSync and its data-efficient followup work SELMA, improving commonsense T2I seems possible and these methods are very simple baselines. I would love to see the authors reasonings on why these methods can or cannot improve commonsense T2I. If they can't, it could further strengthen the necessity of having this benchmark.

---

> ### Author Rebuttal · Authors · 2024-05-31
>
> Thank you for your insightful comments and valuable suggestions.
>
> **Expand T2I models**
>
> As for the closed-sourced SD-3 model, due to limited computation resources, we manually tested 10% examples from our benchmark, and found that the accuracy is around 28%.
>
> As for latent consistency models, we used https://huggingface.co/SimianLuo/LCM_Dreamshaper_v7 model and the GPT-4O evaluation result is: 22.3%. We will include more details in our final version.
>
> **Potential Improvements**
>
> The purpose of our benchmark is to raise the novel problem of commonsense in T2I and to provide an evaluation benchmark. We agree that people can follow our way to collect data and fine-tune T2I models, and such fine-tuning will be helpful to the models. However, we doubt that the core problem lies in the limited expression of CLIP text embeddings, as shown in Sec 3.3 Analysis, and we need a better text embedding for T2I models to really solve this problem.

---

### Decision · Program_Chairs · 2024-07-10

**Decision:**

Accept

**Comment:**

This paper presents Commonsense-T2I, a novel commonsense benchmark for image generation tools. The benchmark contains uncommon prompts such as "A lightbulb without electricity". The authors show that several SOTA models fail to capture key requirements from the prompts (e.g., generate a lit lightbulb on this prompt). The reviewers all praise the work, the idea and its implementation. Relatively minor concerns seem to be addressed in the author-response.